# Effects of Light Quality Treatments during the Grain Filling Period on Yield, Quality, and Fragrance in Fragrant Rice

**Huijia Xie** [1], **Wenjun Xie** [1], **Shenggang Pan** [1,2,3], **Xuwei Liu** [1], **Hua Tian** [1,2,3], **Meiyang Duan** [1,2,3], **Shuli Wang** [1,2,3], **Xiangru Tang** [1,2,3] and **Zhaowen Mo** [1,2,3],*

1    State Key Laboratory for Conservation and Utilization of Subtropical Agro-Bioresources, College of Agriculture, South China Agricultural University, Guangzhou 510642, China; scauxiehj@stu.scau.edu.cn (H.X.); wenjunxie52@163.com (W.X.); panshenggang@scau.edu.cn (S.P.); sumuyulanda@stu.scau.edu.cn (X.L.); tianhua@scau.edu.cn (H.T.); meiyang@scau.edu.cn (M.D.); wangshuli@scau.edu.cn (S.W.); tangxr@scau.edu.cn (X.T.)
2    Scientific Observing and Experimental Station of Crop Cultivation in South China, Ministry of Agriculture and Rural Affairs, Guangzhou 510642, China
3    Guangzhou Key Laboratory for Science and Technology of Fragrant Rice, Guangzhou 510642, China
*    Correspondence: zwmo@scau.edu.cn

**Abstract:** The effect of the light quality on 2-acetyl-1-pyrroline (2AP) during the grain filling period in fragrant rice has rarely been investigated. A pot experiment was carried out with two fragrant rice varieties, Xiangyaxiangzhan and Yuxiangyouzhan, grown under three light treatments, 100% red light (L1), 100% blue light (L2), and compound light (L3), during the grain filling period, and natural light was taken as the control (CK). The yield, quality, and fragrance were investigated. The results showed that light quality treatments significantly decreased the 2AP content in mature grains by 16.67–32.82% but improved the grain yield by 2.70–21.41% compared to CK. The regulation effects of light quality treatments on grain yield and 2AP are linked to yield-related traits, biomass accumulation, antioxidant physiology, and 2AP formation-related physiology. Additionally, light quality treatments decreased the chalky rice percentage and chalkiness, and increased the length-to-width ratio. Overall, light quality treatments during the grain filling period had a positive effect on the grain yield but not on fragrance in fragrant rice.

**Keywords:** 2-acetyl-1-pyrroline; fragrant rice; grain yield; grain quality; light quality

## 1. Introduction

Rice is one of the most popular food crops in the world, and fragrant rice is a specific rice type with high economic value and broad market prospects due to its fragrance and delicacy [1]. Studies have reported that it contains lots of volatiles, of which 2-acetyl-1-pyrroline (2AP) is generally considered to be the most important compound for fragrance in the grain of fragrant rice [2,3]. Studies suggest that plant date [4], salt treatment [5], nutrient (zinc) application [6], and low temperature [7] affect 2AP formation. Besides, fragrant rice has a lower grain yield than non-fragrant rice, and one study has suggested that the fragrance in fragrant rice is related to the reduction of grain yield under salt treatment [8]. It is an intractable scientific issue to improve both grain yield and fragrance in fragrant rice.

Generally, crop management practices are important for rice grain production and quality formation. Previous studies have reported crop management practices, such as application of plant growth regulators [9,10], silicon [11], nitrogen [12], water–nitrogen interaction [13,14], and selenium–silicon interaction [15] effect on both grain yield and fragrance in fragrant rice. Those studies suggested that proline, 1-pyrroline-5-carboxylate (P5C), γ-aminobutyric acid (GABA), as related substances of 2AP synthesis, are highly related to aroma synthesis of fragrant rice. Regarding the growth environment of the



rice plant, light is one of the most important factors associated with grain yield and rice quality [16]. The regulation of grain yield under low light intensity could be associated with changes in gas exchange parameters, dry matter accumulation, and partition [17,18]. For fragrant rice, shading during the grain filling period could lead to yield reduction but improved fragrance, regulating the grain quality [19]. Further, Li et al. [20] reported shading and water stress during the early grain filling stage effect on the grain yield and fragrance of fragrant rice. Therefore, it is generally agreed that low light benefits fragrance accumulation but not grain yield. Moreover, a study indicated that red light and blue light could enhance the activity of nitrate reductase (NR) and the uptake of nitrate in etiolated rice seedlings [21]. In particular, rice resistance to brown spot can be induced by red light [22], and blue light can upregulate the genes of brassinolide in rice seedlings, which is not found in red and far-red light and is beneficial to the bending and unfolding of rice leaves [23]. In addition, Ryo et al. [24] showed that rice plants growing under red light supplemented with blue light could increase leaf total nitrogen content and enhance light saturation and light-limiting photosynthesis. These studies have demonstrated the effects of different light qualities, especially red and blue light, on rice growth. However, the effect of light quality on the fragrance of fragrant rice is not clear. Can changing the light quality induce both grain yield and fragrance of fragrant rice?

The grain filling period is one of the key phases for grain yield, quality, and fragrance formation. Therefore, in the present study, three different light quality treatments during the grain filling period were employed to investigate the light quality effect on the grain yield, quality, and fragrance of fragrant rice. The objective of this study is to try to assess what light quality changes fragrance most and the possible relationship between grain yield and fragrance accumulation under different qualitative conditions.

## 2. Materials and Methods

### 2.1. Plant Materials and Description of Experiment

The seeds of two fragrant rice varieties (Xiangyaxiangzhan and Yuxiangyouzhan) used in this study were collected from the College of Agriculture, South China Agricultural University. Xiangyaxiangzhan and Yuxiangyouzhan are inbred long grain rice varieties with a growth period of 112–114 days and 126–128 days, respectively. These two rice varieties are widely cultivated in the local region.

A pot experiment was performed from March to July 2018 at the Experimental Farm of College of Agriculture, South China Agricultural University, Guangzhou, China. The experimental soil was sandy loam containing organic matter 36.95 g kg$^{-1}$, total nitrogen 1.94 g kg$^{-1}$, total phosphorous 1.32 g kg$^{-1}$, and total potassium 22.96 g kg$^{-1}$, with a pH of 6.40.

Rice seedlings with three leaves were transplanted at three seedlings per hill and five hills per pot. The pot size was 32 cm in diameter and 24 cm in height, containing 12 kg air-dried soil. The Norway compound fertilizer (N:P$_2$O$_5$:K$_2$O = 15:15:15) was applied at 7 g per pot with a ratio of base fertilizer:tiller fertilizer = 50:50. The water layer was maintained during the rice-growing season, and chemical pesticides and herbicides were used to avoid yield loss caused by disease, insects, and weeds.

### 2.2. Experiment Design

The experiment was arranged in randomized complete block design. The different light quality treatments were applied from 10 to 30 June 2018, that is, from R5 to R9 stage, as described by Counce et al. [25]. The light quality treatments included additionally supplying 100% red light (L1), 100% blue light (L2), and compound light (L3, red light:blue light:white light = 1:1:1). The rice plant growth under natural light conditions was taken as the control (CK). The LED lamps that supplied light were reported previously by Fang et al. [26].

### 2.3. Sampling and Measurements

At 15 days after the treatments (15dAT) and maturity stages (MS), the leaves and grain samples were harvested for measurement of 2AP content in the grain and the physiological parameters. The leaves and grain samples were immediately frozen by liquid nitrogen and stored at $-80\ ^{\circ}$C for the determination of $\gamma$-aminobutyric acid (GABA) content, 1-pyrroline-5-carboxylate (P5C) content, proline content, antioxidant enzyme activity, and malondialdehyde (MDA) content. Another set of the grain sample was stored at $-20\ ^{\circ}$C for the determination of 2AP content.

#### 2.3.1. Determination of 2AP Content in Grain

The grain sample was ground to powder, then 2.0 g of powder was weighed for measurement of 2AP content. The measurement of 2AP content was carried out by using the GCMS-QP 2010 plus (Shimadzu Corporation, Kyoto, Japan) method [13]. The 2AP content was expressed as $ug \cdot g^{-1}$ dry weight (DW).

#### 2.3.2. Determination of Proline, P5C, and GABA Content

The proline content was measured according to the method of Bates et al. [27]. Fresh plant tissue (0.3 g) was extracted in 3% sulfosalicylic acid (5 mL) and kept in boiling water for 10 min, then cooled. A total of 1 mL of supernatant was mixed successively with 1 mL of glacial acetic acid and 1 mL of 2.5% ninhydrin reagent and then kept in boiling water for 30 min. The reaction mixture was extracted by 4 mL toluene and then standing stratification. The 1 mL extract was centrifuged at 4000 rpm for 5 min. The absorbance was recorded at 530 nm. The proline content was expressed as $ug \cdot g^{-1}$ fresh weight (FW). The determination of P5C content was conducted according to the method of Miller et al. [28]. Fresh plant tissue (0.5 g) was weighed and homogenized in 6 mL sulfosalicylic acid. The homogenate was centrifuged at 10,000 rpm for 10 min. The supernatant (1.35 mL) was mixed with 1.5 mL of 10% trichloroacetic acid and 0.15 mL of 2-amino benzaldehyde. The sample was kept at room temperature for 25 min and then centrifuged at 10,000 rpm for 10 min. After centrifugation, the absorbance was measured at 440 nm. The P5C content was calculated by using the molar extinction coefficient of P5C ($2.58\ mmol \cdot cm^{-1}$) according to Mezl and Knox [29]. The P5C content was expressed in $\mu mol\ g^{-1}$ FW. The determination of GABA content was conducted according to the method of Zhao et al. [30]. After reaction, the absorbance was measured at 645 nm. The GABA content was expressed as $mg \cdot g^{-1} \cdot FW$.

#### 2.3.3. Antioxidant Enzyme Activity and MDA Content

The antioxidant enzyme activity was conducted according to the method of Kong et al. [31] and Li et al. [32]. The fresh plant tissue (0.3 g) was weighed and homogenized in 5 mL of $50\ mmol \cdot L^{-1}$ phosphate buffer solution (PBS, pH = 7.8), and then centrifuged at 8000 rpm for 15 min. The supernatant was used for the measurement of antioxidant enzyme activity and MDA content. Superoxide dismutase (SOD) was measured using the nitro blue tetrazolium (NBT) method. After reaction, the absorbance was measured at 560 nm. One unit of SOD activity was defined as 50% inhibition of the color reaction. The SOD activity was expressed as $U \cdot g^{-1} \cdot FW$. For the determination of peroxidase (POD), after adding the reaction solution, the absorbance was measured at 470 nm for 2 min and was recorded every 30 s. One unit of POD activity was defined as an absorbance increase of $0.01\ min^{-1}$. The POD activity was expressed as $U \cdot g^{-1} \cdot min^{-1} \cdot FW$. For the catalase (CAT) activity, the enzyme extract (0.02 mL) was reacted with 0.12 mL of $50\ mmol \cdot L^{-1}\ H_2O_2$ for 2 min. Then, we added 0.4 mL of saturated sodium chloride solution and 1 mL of $50\ mmol \cdot L^{-1}$ ammonium molybdate and homogenized. After 10 min, the absorbance was measured at 405 nm. CAT activity was calculated according to the standard curve. One unit of CAT activity was defined as an absorbance decrease of $0.01\ min^{-1}$. CAT activity was expressed as $mmol \cdot min^{-1} \cdot g^{-1} \cdot FW$. For the determination of MDA, 1.5 mL of enzyme extract and 2 mL of 0.5% thiobarbituric acid solution which is soluble in 5% trichloroacetic acid solution were added into a 5-mL centrifuge cube and homogenized fully, then kept

in boiling water for 30 min. Then, the centrifuge cube was removed immediately and cooled to room temperature. The homogenate was centrifuged at 3000 rpm for 15 min. The absorbance was measured at 532 nm, 600 nm, and 450 nm. The MDA was expressed as $\mu mol \cdot g^{-1} \cdot FW$.

### 2.3.4. Determination of Dry Weight, Grain Yield

At the maturity stage, the rice grain was harvested from six pots. The number of panicles per pot was counted. After natural air drying, the filled grain number per panicle, total grain per panicle, 1000-grain weight, and grain yield were recorded; then, the filled grain percentage was calculated. The plant samples were separated into leaves, stems, and panicles and fixed at 105 °C for 30 min. Then, they were oven-dried at 80 °C to a constant weight for the determination of dry weight. The harvest index was calculated as grain yield divided by total dry weight.

### 2.3.5. Determination of Grain Quality

The grain quality was determined according to the method of Mo et al. [19]. The grain samples after natural drying were stored at room temperature for 3 months. Milling quality, including brown rice rate, milled rice rate, and head milled rice rate, was measured. The appearance quality, including chalky rice rate, chalkiness, and length-to-width ratio, was recorded. The protein content, amylose content, and alkali value were measured by using an Infratec 1241 grain analyzer (Foss Tecator Co., Ltd., Hillerod, Denmark).

### 2.4. Statistical Analysis

Microsoft Office 2013 was used for data collection and plotting. Analyses of variance (ANOVAs) were performed in accordance with the linear model procedure of Statistix Statistical Software version 8 (Statistix 8, Analytical, Tallahassee, FL, USA). Comparisons of means between different treatments were performed using the least significant difference (LSD) test at a 5% probability level.

## 3. Results

### 3.1. AP Content

Variety (V) significantly affected 2AP content in grains at 15 days after treatment. Significant light treatment (T) and V×T effect on 2AP content in grains was observed (Table 1). Compared with CK, a significant reduction in the 2AP content in grains of Xiangyaxiangzhan at 15 d AT for L1, L2, and L3 treatments was observed by 14.83%, 9.51%, and 17.49% (Figure 1A), respectively. The 2AP content in grains of Yuxiangyouzhan at 15 d AT for L2 treatments decreased significantly by 22.32% (Figure 1A). At the maturity stage, compared with CK, the 2AP content in grains of Xiangyaxiangzhan decreased by 31.82%, 27.27%, and 27.27% for L1, L2, and L3 treatments, respectively (Figure 1B). The L2 and L3 treatments decreased the 2AP content the least in Xiangyaxiangzhan. The L1, L2, and L3 treatments significantly reduced the 2AP content in the grain of Yuxiangyouzhan at MS by 16.67%, 16.67%, and 16.67%, respectively (Figure 1B).

### 3.2. Proline, P5C, and GABA Content in Leaves and Grain

Variety (V) significantly affected the proline content in grains. Significant light quality treatment (T) and V×T effect on 2AP content in grains and leaves were observed (Table 1). For Xiangyaxiangzhan, all light quality treatments significantly decreased the proline content in the grains at 15 d AT, whereas no significant difference was noted in the proline content in the grain at MS (Figure 2B). The L2 treatment significantly increased the proline content in the leaves at 15 d AT by 28.10% (Figure 2C). The L1 and L2 treatments significantly increased the proline content in the leaves at MS by 31.07% and 13.30%, respectively, while the L3 treatment significantly reduced the proline content in the leaves at MS (Figure 2D). For Yuxiangyouzhan, the L2 treatment significantly reduced the proline content in grains at 15 d AT by 28.97% (Figure 2A). However, the L3 treatment significantly

increased the proline content in grains at 15 d AT by 54.01% (Figure 2A). All light quality treatments reduced the proline content in the grains at MS by 10.48–39.04% (Figure 2B). The L2 treatment and L1 treatment significantly decreased the proline content in leaves at 15 d AT and MS, respectively (Figure 2C,D).

**Table 1.** ANOVA of the investigated parameters.

| Parameters | F Values | | |
|---|---|---|---|
| | Variety (V) | Treatment (T) | V×T |
| 2AP content in grains at 15 d AT | 1614.85 ** | 7.65 ** | 4.01 * |
| 2AP content in grains at MS | 5.92 ns | 19.21 ** | 4.46 * |
| P5C content in grains at 15 d AT | 21.43 * | 44.45 ** | 11.00 ** |
| P5C content in grains at MS | 307.20 ** | 163.31 ** | 19.36 ** |
| Proline content in grains at 15 d AT | 78.69 ** | 284.52 ** | 379.34 ** |
| Proline content in grains at MS | 60.24 ** | 13.01 ** | 16.75 ** |
| GABA content in grains at 15 d AT | 160.84 ** | 68.39 ** | 8.68 ** |
| GABA content in grains at MS | 0.05 ns | 11.00 ** | 2.46 ns |
| P5C content in leaves at 15 d AT | 20.52 * | 10.24 ** | 16.57 ** |
| P5C content in leaves at MS | 27.32 * | 7.30 ** | 12.81 ** |
| Proline content in leaves at 15 d AT | 4.55 ns | 4.39 * | 14.77 ** |
| Proline content in leaves at MS | 0.00 ns | 19.47 ** | 52.53 ** |
| GABA content in leaves at 15 d AT | 6.61 ns | 2.74 ns | 5.84 ** |
| GABA content in leaves at MS | 2.08 ns | 3.82 * | 6.09 ** |
| Grain yield | 1.22 ns | 5.52 ** | 1.24 ns |
| Panicle number per pot | 6.69 ns | 0.63 ns | 0.21 ns |
| Grain number per panicle | 34.91 ** | 6.25 ** | 2.97 ns |
| Filled grain percentage | 37.37 ** | 2.46 ns | 0.18 ns |
| 1000-grain weight | 73.91 ** | 2.00 ns | 1.38 ns |
| Stem and leaf dry weight | 0.36 ns | 2.74 ns | 1.71 ns |
| Total dry weight | 2.06 ns | 0.44 ns | 2.69 ns |
| Harvest index | 0.12 ns | 5.01 * | 0.94 ns |
| SOD activity in leaves at 15 d AT | 21.52 * | 5.14 ** | 4.48 * |
| SOD activity in leaves at MS | 146.87 ** | 85.22 ** | 82.41 ** |
| POD activity in leaves at 15 d AT | 75.09 ** | 12.74 ** | 16.96 ** |
| POD activity in leaves at MS | 17.85 * | 12.86 ** | 7.47 ** |
| CAT activity in leaves at 15 d AT | 89.15 ** | 72.89 ** | 26.29 ** |
| CAT activity in leaves at MS | 602.45 ** | 128.04 ** | 191.53 ** |
| MDA content in leaves at 15 d AT | 1.59 ns | 10.07 ** | 9.62 ** |
| MDA content in leaves at MS | 187.36 ** | 49.06 ** | 17.40 ** |
| Brown rice rate | 451.20 * | 28.42 ** | 34.15 ** |
| Milled rice rate | 2.38 ns | 0.83 ns | 0.36 ns |
| Head rice rate | 105.56 ns | 4.54 ns | 1.19 ns |
| Chalk rice percentage | 2361.90 ** | 66.05 ** | 14.61 ** |
| Chalkiness | 1536.13 ** | 75.36 ** | 14.45 ** |
| Length-to-width ratio | 16418.8 ** | 48.53 ** | 15.31 ** |
| Protein | 346.69 ** | 30.50 ** | 8.35 ** |
| Amylose | 7224.03 ** | 1611.82 ** | 1731.09 ** |
| Alkali value | 1225.00 ** | 13.53 ** | 24.73 ** |

ns, not significant; *: significant at $p < 0.05$ level; **: significant at $p < 0.01$ level. CK, natural light condition; L1, 100% red light; L2, 100% blue light; L3, compound light (L3, red light:blue light:white light = 1:1:1); 15 d AT, 15 days after treatment; MS: maturity stage; 2AP, 2-acetyl-1-pyrroline; P5C, 1-pyrroline-5-carboxylate; GABA, γ-aminobutyric acid; SOD, superoxide dismutase; POD, peroxidase; CAT, catalase; MDA, malondialdehyde.

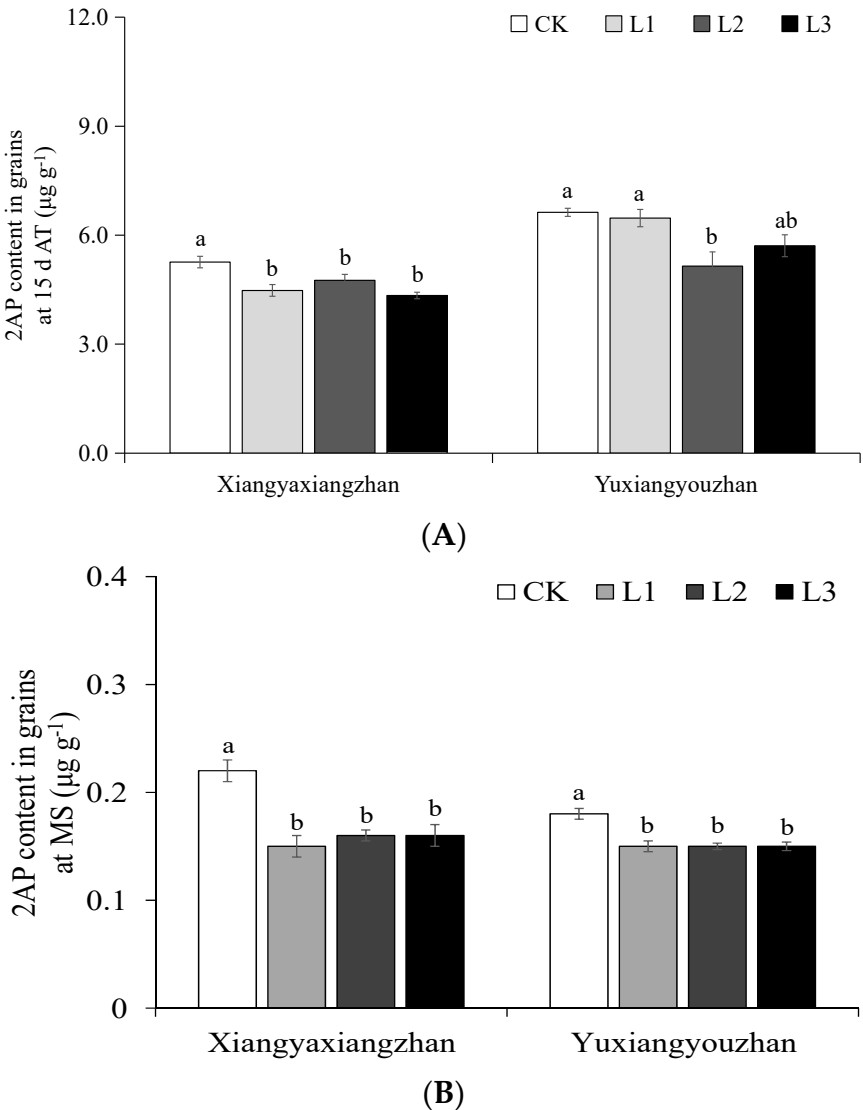

**Figure 1.** Effect of light quality on 2AP content in grains at 15 dAT (**A**) and at MS (**B**). Vertical bars represent the mean value. Capped bars above represent the standard error of three replicates. Means sharing a different lower-case letter differ significantly at *p* < 0.05 according to the least significant difference (LSD) test. CK, natural light condition; L1, 100% red light; L2, 100% blue light; L3, compound light (L3, red light:blue light:white light = 1:1:1); 15 d AT, 15 days after treatment; MS, maturity stage; 2AP, 2-acetyl-1-pyrroline.

Variety (V), light quality treatment (T), and V×T significantly affected P5C content in grains and leaves (Table 1). Compared with CK, the P5C content in grains in Xiangyaxiangzhan and Yuxiangyouzhan was significantly decreased at 15 d AT and MS under all the light quality treatment (Figure 3A,B). Compared with CK, the P5C content in leaves in Xiangyaxiangzhan at 15 d AT was significantly increased at L1 and L3 treatments, but the P5C content in leaves in Yuxiangyouzhan at 15 d AT was significantly decreased under L2 and L3 treatments (Figure 3C). At MS, the P5C content in leaves in Xiangyaxiangzhan was increased under L1 and L2 treatments, whereas increments in the P5C content in leaves in Yuxiangyouzhan were observed for L1 and L3 treatments as compared to CK (Figure 3D).

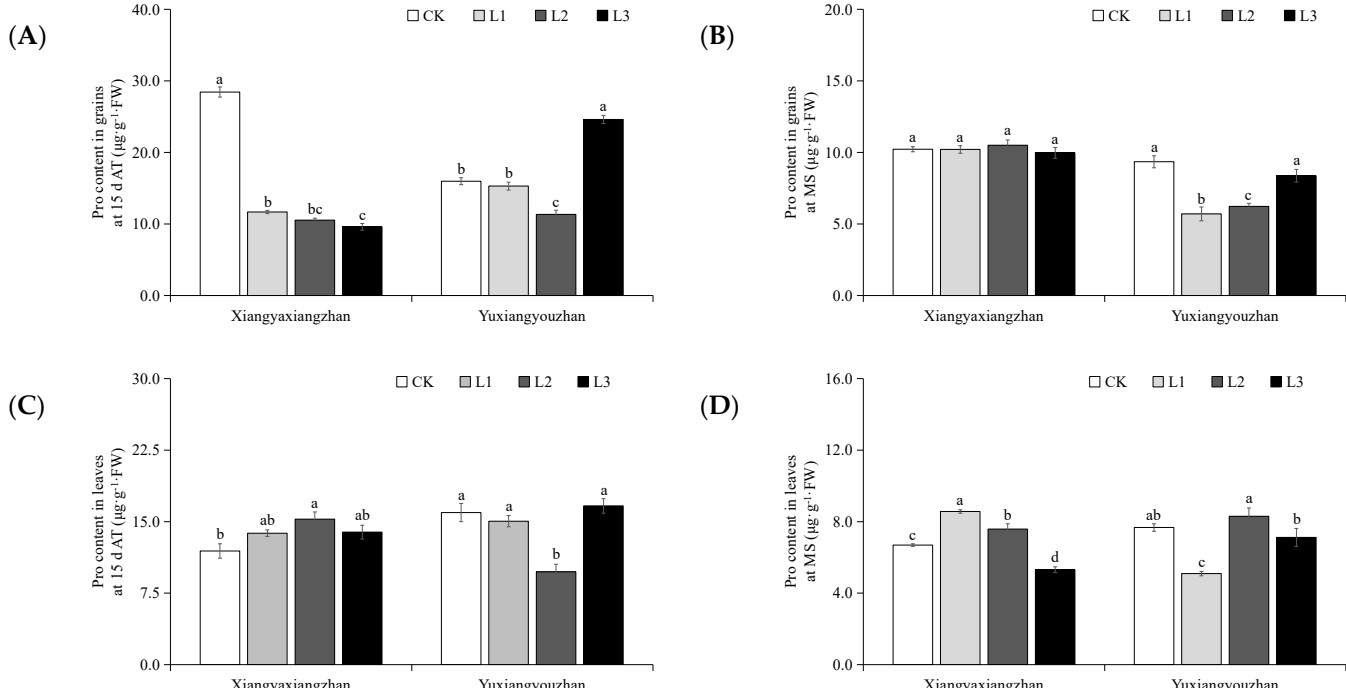

**Figure 2.** Effect of light quality on proline content in grains at 15 dAT (**A**) and at MS (**B**) and leaves at 15 dAT (**C**) and at MS (**D**). Vertical bars present the mean value. Capped bars above represent the standard error of three replicates. Means sharing a different lower-case letter differ significantly at *p* < 0.05 according to the least significant difference (LSD) test. CK, natural light condition; L1, 100% red light; L2, 100% blue light; L3, compound light (red light:blue light:white light = 1:1:1); 15 d AT, 15 days after treatment; MS, maturity stage; Pro, proline. FW, fresh weight.

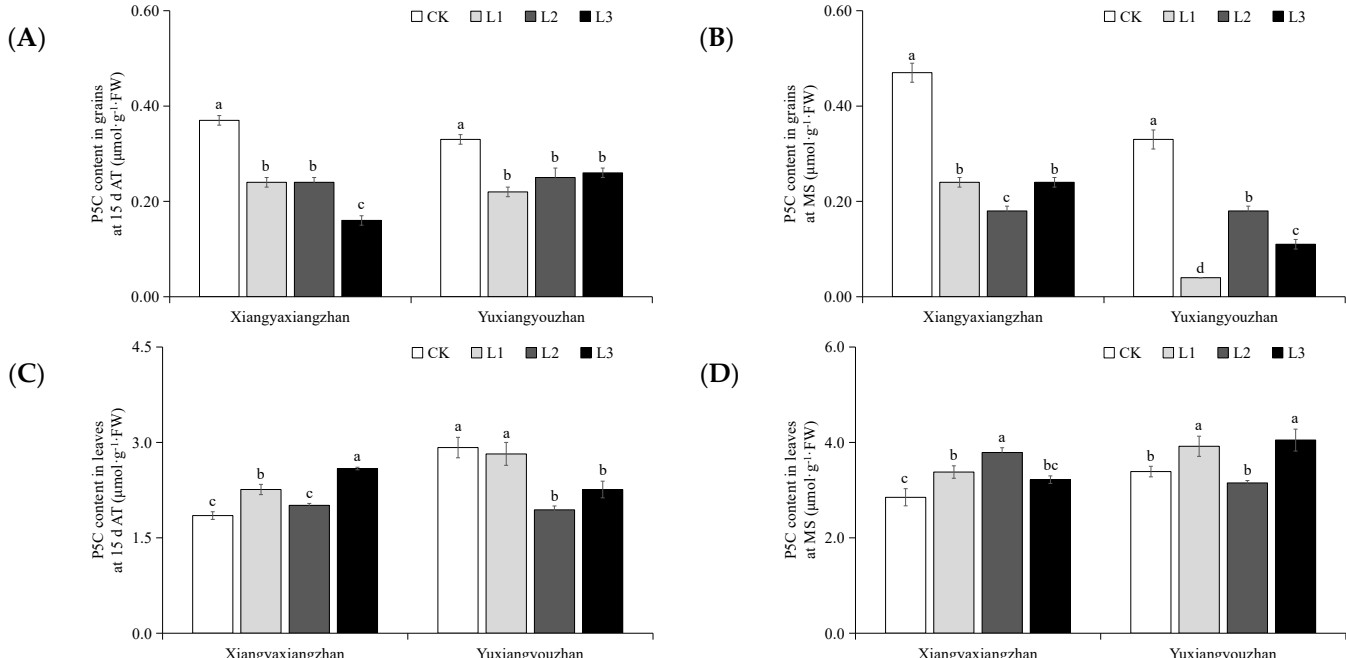

**Figure 3.** Effect of light quality on P5C content in grains at 15 dAT (**A**) and at MS (**B**) and leaves at 15 dAT (**C**) and at MS (**D**). Vertical bars represent the mean value. Capped bars above represent the standard error of three replicates. Means sharing a different lower-case letter differ significantly at *p* < 0.05 according to the least significant difference (LSD) test. CK, natural light condition; L1, 100% red light; L2, 100% blue light; L3, compound light (red light:blue light:white light = 1:1:1); 15 d AT, 15 days after treatment; MS, maturity stage; P5C, 1-pyrroline-5-carboxylate. FW, fresh weight.

Variety (V) significantly affected the GABA content in grains at 15d AT. Light quality treatment (T) significantly affected the GABA content in grains at 15 d AT and at MS and in leaves at 15 d AT. V×T significantly affected the GABA content in grains at 15 d AT and in leaves at 15 d AT and at MS (Table 1). For Xiangyaxiangzhan, compared with CK, all light quality treatments significantly decreased the GABA content in grains at 15 d AT (Figure 4A), whereas L2 and L3 treatments increased the GABA content in grains at MS (Figure 4B). The L2 treatment significantly decreased the GABA content in leaves in Xiangyaxiangzhan at 15 d AT (Figure 4C), but L1 and L2 treatments significantly increased the GABA content in leaves in Xiangyaxiangzhan at MS (Figure 4D). For Yuxiangyouzhan, compared with CK, all light quality treatments significantly decreased the GABA content in grains at 15 d AT (Figure 4A), while L1 treatment significantly decreased the GABA content in grains at MS (Figure 4B). No significant difference was noted in the GABA content in the leaves in Yuxiangyouzhan (Figure 4C,D).

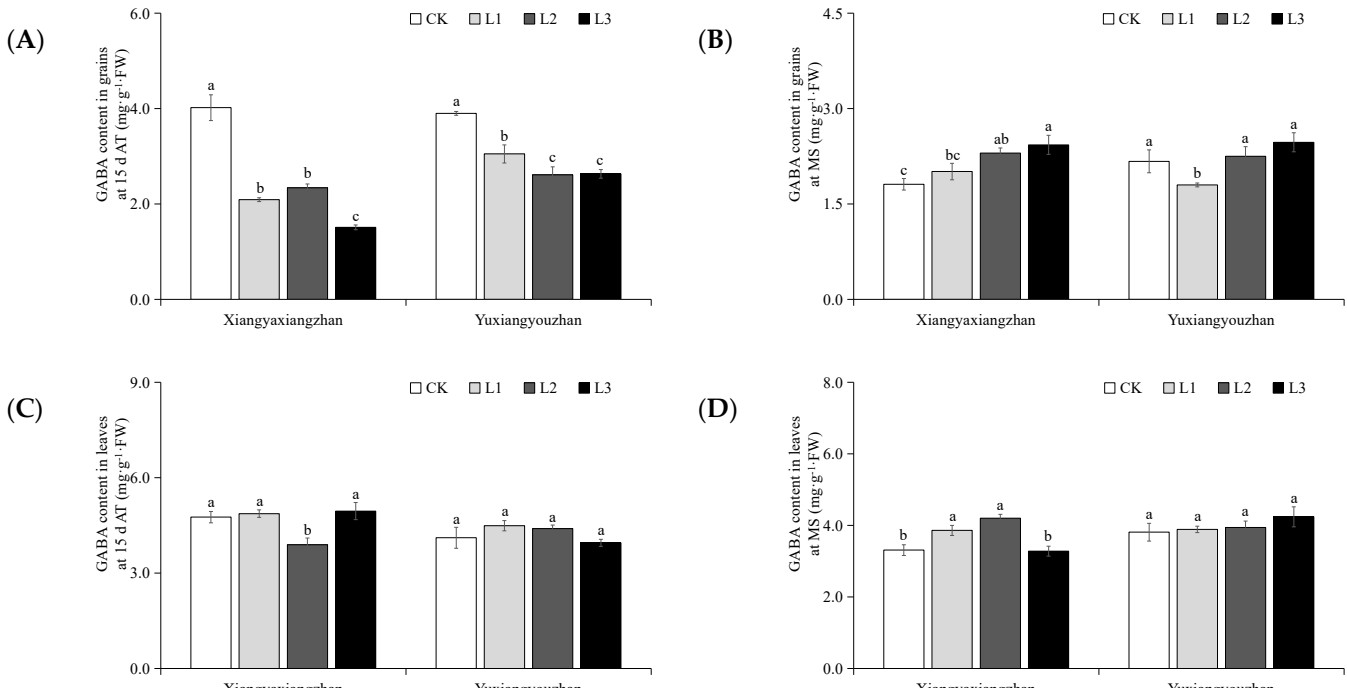

**Figure 4.** Effect of light quality on GABA content in grains at 15 dAT (**A**) and at MS (**B**) and leaves at 15 dAT (**C**) and at MS (**D**). Vertical bars present the mean value. Capped bars above represent the standard error of three replicates. Means sharing a different lower-case letter differ significantly at *p* < 0.05 according to the least significant difference (LSD) test. CK, natural light condition; L1, 100% red light; L2, 100% blue light; L3, compound light (L3, red light:blue light:white light = 1:1:1); 15 d AT, 15 days after treatment; MS, maturity stage; GABA, γ-aminobutyric acid; FW, fresh weight.

### 3.3. Yield, Yield-Related Traits, and Biomass

Variety (V) significantly affected grain number per panicle, filled grain percentage, and 1000-grain weight. A significant light quality treatment (T) effect on grain yield, grain number per panicle, and harvest index was observed (Table 1). For Xiangyaxiangzhan, compared with CK, L1 and L3 treatments significantly increased the grain number per panicle and grain yield, and all light quality treatments significantly improved the filled grain percentage. The grain yield increment of Xiangyaxiangzhan under L3 treatment was the highest. For Yuxiangyouzhan, compared with CK, all light quality treatments significantly enhanced the grain yield (up to 19.46–21.41%). L2 and L3 treatments significantly increased the grain number per panicle in Yuxiangyouzhan. The L1 and L3 treatments significantly improved the harvest index in Yuxiangyouzhan (Table 2).

**Table 2.** Effect of light quality treatment on grain yield, yield-related traits, biomass, and harvest index.

| Variety | Treatment | Panicle Number per Pot | Grains per Panicle | Filled Grain Percentage (%) | 1000-Grain Weight (g) | Grain Yield (g pot$^{-1}$) | Stem and Leaves Dry Weight (g pot$^{-1}$) | Total Dry Weight (g pot$^{-1}$) | Harvest Index |
|---|---|---|---|---|---|---|---|---|---|
| Xiangyaxiangzhan | CK | 15.25 ± 0.25 a | 87.61 ± 4.02 b | 87.20 ± 2.44 b | 20.68 ± 0.19 a | 24.84 ± 1.58 c | 45.14 ± 2.71 a | 69.98 ± 2.25 a | 0.36 ± 0.03 a |
| | L1 | 16.00 ± 0.41 a | 98.95 ± 1.71 a | 92.56 ± 0.40 a | 19.85 ± 0.30 a | 28.53 ± 0.45 ab | 40.95 ± 3.80 a | 69.49 ± 3.81 a | 0.41 ± 0.03 a |
| | L2 | 16.00 ± 0.98 a | 87.71 ± 1.69 b | 91.76 ± 0.73 a | 20.20 ± 0.52 a | 25.51 ± 0.63 bc | 42.32 ± 1.69 a | 67.83 ± 1.67 a | 0.38 ± 0.01 a |
| | L3 | 15.50 ± 0.50 a | 102.34 ± 3.10 a | 92.97 ± 1.49 a | 19.70 ± 0.29 a | 29.46 ± 1.36 a | 43.77 ± 1.65 a | 73.23 ± 0.29 a | 0.40 ± 0.02 a |
| Yuxiangyouzhan | CK | 16.58 ± 0.55 a | 100.09 ± 3.96 b | 67.92 ± 4.08 a | 22.02 ± 0.12 a | 25.03 ± 1.94 b | 52.14 ± 3.98 a | 77.17 ± 3.81 a | 0.33 ± 0.03 b |
| | L1 | 17.25 ± 0.48 a | 105.26 ± 3.92 ab | 76.37 ± 4.58 a | 22.28 ± 0.20 a | 29.90 ± 2.15 a | 43.45 ± 3.85 ab | 73.34 ± 2.13 a | 0.41 ± 0.04 a |
| | L2 | 16.50 ± 0.29 a | 113.48 ± 3.91 a | 73.19 ± 5.63 a | 21.95 ± 0.24 a | 30.39 ± 2.43 a | 45.44 ± 4.39 ab | 75.83 ± 4.38 a | 0.40 ± 0.03 ab |
| | L3 | 16.50 ± 0.65 a | 114.64 ± 3.40 a | 73.12 ± 3.53 a | 21.68 ± 0.21 a | 30.38 ± 1.91 a | 38.06 ± 3.39 b | 68.44 ± 3.35 a | 0.45 ± 0.03 a |

Means in the same column for the same variety followed by different lower-case letters differ significantly at *p* < 0.05 according to LSD test.
CK, natural light condition; L1, 100% red light; L2, 100% blue light; L3, compound light (L3, red light:blue light:white light = 1:1:1).

### 3.4. Antioxidant Enzyme Activity and MDA Content in Leaves

Significant variety (V), light quality treatment (T), and V×T effect on the antioxidant enzyme activity in leaves and MDA content at MS in leaves were detected (Table 1). Compared with CK, the SOD activity in leaves in Xiangyaxiangzhan and Yuxiangyouzhan at 15 d AT significantly increased with L1 and L2 treatment, respectively. At MS, the L1 and L3 treatments significantly reduced the SOD activity in leaves in Yuxiangyouzhan (Figure 5A,B). Compared with CK, L1, and L2 treatment significantly decreased POD activity in leaves in Xiangyaxiangzhan at 15 d AT. L1 treatment significantly improved POD activity in leaves in Xiangyaxiangzhan at MS, while L3 treatment significantly reduced POD activity in leaves in Xiangyaxiangzhan at MS. POD activity in leaves in Yuxiangyouzhan at 15 d AT was significantly reduced under L1 and L3 treatments, but POD activity in leaves in Yuxiangyouzhan was significantly increased under L2 treatment. At MS, POD activity in leaves in Yuxiangyouzhan significantly reduced under L2 treatment (Figure 5C,D). Compared with CK, CAT activity in leaves in Xiangyaxiangzhan at 15 d AT significantly increased under L1 treatment but reduced under L2 treatment. Besides, L1 and L2 treatment significantly increased CAT activity in leaves in Xiangyaxiangzhan at MS. For Yuxiangyouzhan, CAT activity in leaves at 15 d AT significantly increased under L1 and L3 treatments but reduced under L2 treatment. All the light quality treatments significantly reduced CAT activity in leaves at MS (Figure 5E,F). For Xiangyaxiangzhan, compared with CK, MDA content significantly increased under all light quality treatments at both 15 d AT and MS. For Yuxiangyouzhan, compared with CK, the MDA content was significantly increased by L1 and L2 treatments at both 15 d AT and MS. L3 treatment significantly decreased the MDA content in leaves at 15 d AT in Yuxiangyouzhan (Figure 5G,H).

### 3.5. Grain Quality

Variety (V), light quality treatment (T), and V×T significantly affected the grain quality except for milled rice rate and head rice rate (Table 1). For Xiangyaxiangzhan, compared with CK, L3 treatment significantly reduced head rice rate. L2 and L3 treatments significantly reduced the alkali value. All light quality treatments significantly reduced brown rice rate, chalk rice percentage, chalkiness, protein content, and amylose content but increased the length-to-width ratio. For Yuxiangyouzhan, compared with CK, brown rice rate significantly increased under L1 treatment. L2 treatment significantly increased the amylose content. The protein content was significantly decreased under L2 and L3 treatments. The chalk rice percentage and chalkiness were significantly decreased under all light quality treatments. The length-to-width ratio and alkali value were significantly increased by all light quality treatments (Table 3).

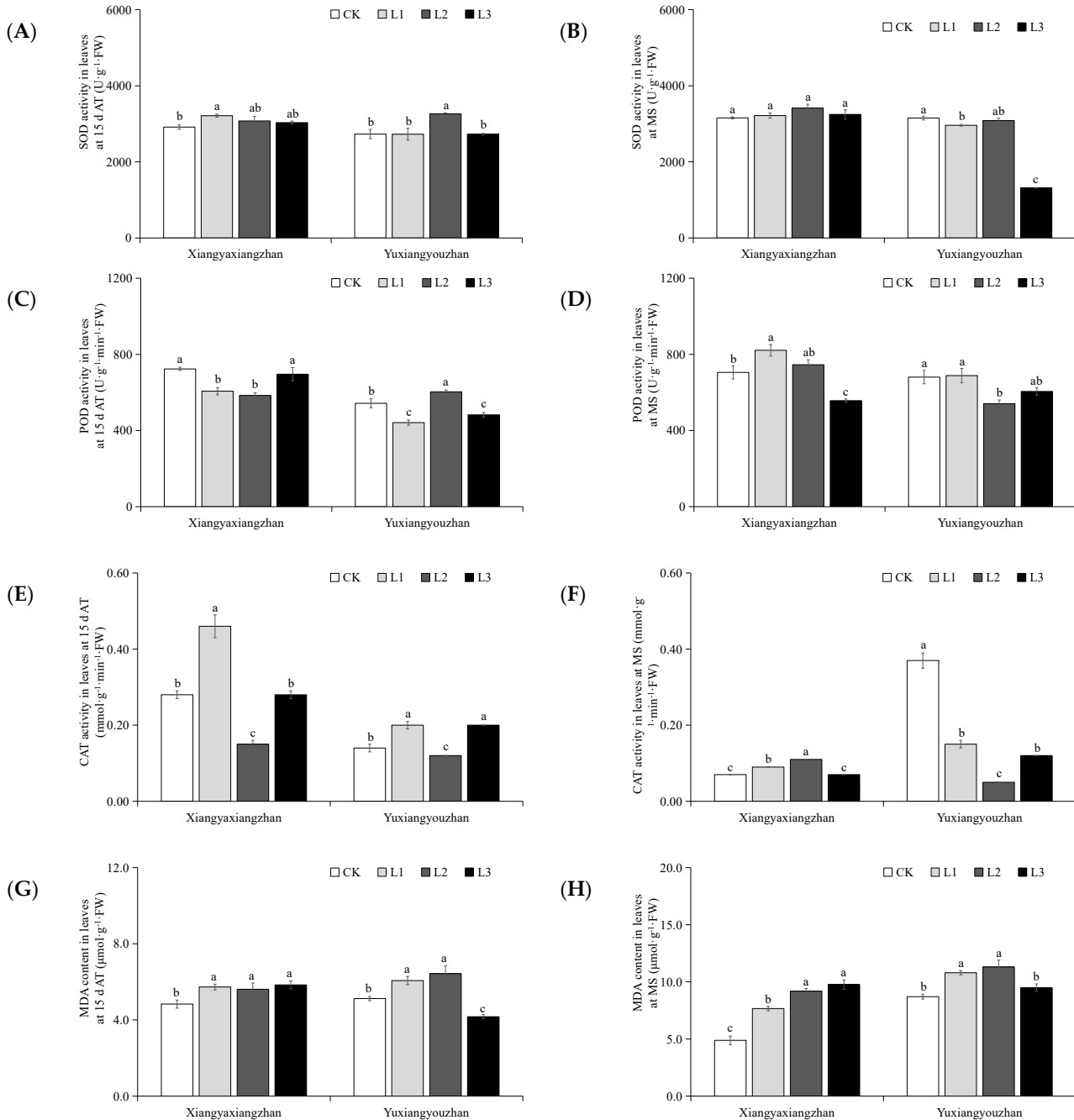

**Figure 5.** Effect of light quality on SOD activity in leaves at 15 dAT (**A**) and at MS (**B**), POD activity in leaves at 15 dAT (**C**) and at MS (**D**), CAT activity in leaves at 15 dAT (**E**), and at MS (**F**), and MDA content in leaves at 15 dAT (**G**) and at MS (**H**). Vertical bars represent the mean value. Capped bars above represent the standard error of three replicates. Means sharing a different lower-case letter differ significantly at *p* < 0.05 according to the least significant difference (LSD) test. CK: natural light condition; L1, 100% red light; L2, 100% blue light; L3, compound light (L3, red light:blue light:white light = 1:1:1); 15 d AT, 15 days after treatment; MS, maturity stage. SOD, superoxide dismutase; POD, peroxidase; CAT, catalase; MDA, malondialdehyde; FW, fresh weight.

**Table 3.** Effect of light quality treatment on grain quality parameters.

| Variety | Treatment | Brown Rice Rate (%) | Milled Rice Rate (%) | Head Rice Rate (%) | Chalky Rice Percentage (%) | Chalkiness (%) | Length-to-Width Ratio | Protein Content (%) | Amylose Content (%) | Alkali Value |
|---|---|---|---|---|---|---|---|---|---|---|
| Xiangyaxiangzhan | CK | 78.98 ± 0.00 a | 71.04 ± 0.20 a | 50.71 ± 0.00 a | 22.40 ± 1.47 a | 11.19 ± 0.57 a | 2.72 ± 0.03 c | 7.93 ± 0.03 a | 20.70 ± 0.15 a | 6.27 ± 0.03 a |
| | L1 | 77.36 ± 0.04 b | 70.08 ± 0.81 a | 46.16 ± 1.81 ab | 4.80 ± 0.37 b | 1.70 ± 0.11 c | 3.06 ± 0.01 a | 7.37 ± 0.03 b | 16.97 ± 0.09 b | 6.20 ± 0.00 ab |
| | L2 | 77.52 ± 0.28 b | 69.81 ± 1.28 a | 45.65 ± 0.90 ab | 3.80 ± 0.37 c | 1.31 ± 0.08 c | 2.91 ± 0.03 b | 7.13 ± 0.07 c | 17.03 ± 0.12 b | 6.17 ± 0.03 b |
| | L3 | 77.40 ± 0.02 b | 70.91 ± 1.94 a | 41.78 ± 1.85 b | 17.00 ± 0.84 c | 7.94 ± 0.58 b | 3.07 ± 0.03 a | 7.47 ± 0.03 b | 17.03 ± 0.09 b | 6.03 ± 0.03 c |
| Yuxiangyouzhan | CK | 79.31 ± 0.08 b | 73.67 ± 3.81 a | 72.30 ± 4.49 a | 76.59 ± 1.10 a | 37.80 ± 0.82 a | 1.76 ± 0.01 c | 6.60 ± 0.06 a | 26.50 ± 0.00 b | 5.97 ± 0.09 b |
| | L1 | 79.67 ± 0.07 a | 72.40 ± 2.50 a | 71.00 ± 2.35 a | 51.26 ± 2.19 c | 22.13 ± 1.08 c | 1.92 ± 0.03 a | 6.47 ± 0.03 ab | 26.53 ± 0.03 ab | 6.60 ± 0.06 a |
| | L2 | 79.31 ± 0.04 b | 71.35 ± 0.77 a | 69.28 ± 0.60 a | 58.46 ± 2.13 b | 26.64 ± 0.96 b | 1.94 ± 0.04 a | 6.30 ± 0.10 b | 26.60 ± 0.00 a | 6.73 ± 0.07 a |
| | L3 | 79.14 ± 0.01 b | 71.33 ± 1.21 a | 69.43 ± 1.20 a | 52.56 ± 2.70 bc | 23.99 ± 1.82 bc | 1.85 ± 0.01 b | 6.23 ± 0.09 b | 26.57 ± 0.03 ab | 6.53 ± 0.07 a |

Means in the same column for the same variety followed by different lower-case letters differ significantly at $p < 0.05$ according to LSD test; L1, 100% red light; L2, 100% blue light; L3, compound light (L3, red light:blue light:white light = 1:1:1).3.6. Correlation analyses.

The 2AP content in grains at MS positively correlated with the P5C content in grains at 15 d AT ($r^2$ = 0.6714, $p < 0.05$), the P5C content in grains at MS ($r^2$ = 0.7846, $p < 0.01$), and the GABA content in grains at 15d AT ($r^2$ = 0.5222, $p < 0.05$) (Figure 6A,B). Grain yield positively correlated with grain number per panicle ($r^2$ = 0.7285, $p < 0.01$) and harvest index ($r^2$ = 0.7632, $p < 0.01$) (Figure 7A,B).

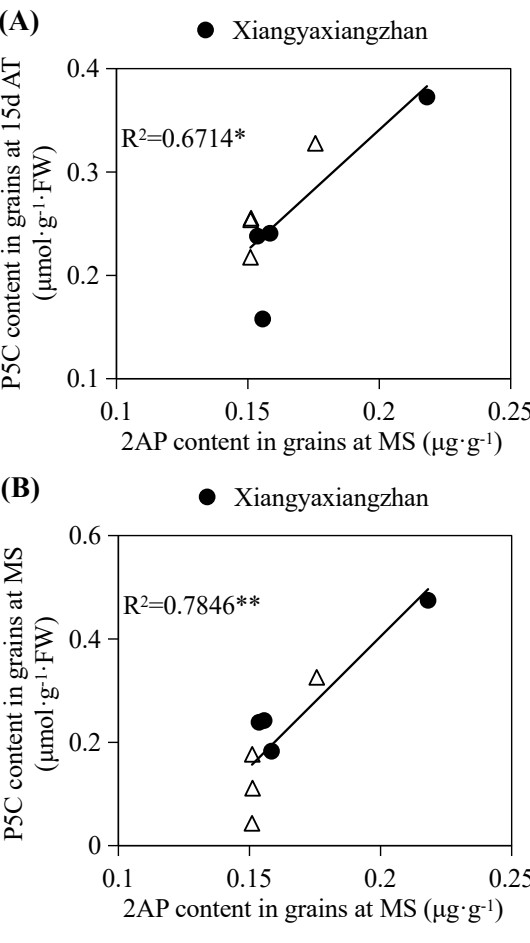

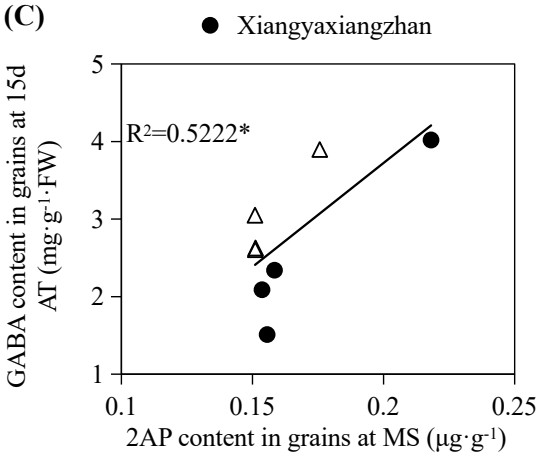

**Figure 6.** Correlation analyses between 2AP content and P5C content in grains at 15 dAT (**A**) and at MS (**B**) and GABA content in grains at 15 d AT (**C**) of fragrant rice. *, significant at $p < 0.05$ level; **: significant at $p < 0.01$ level; 15 d AT, 15 days after treatment; MS, maturity stage. 2AP, 2-acetyl-1-pyrroline; P5C, 1-pyrroline-5-carboxylate; GABA, γ-aminobutyric acid; FW, fresh weight.

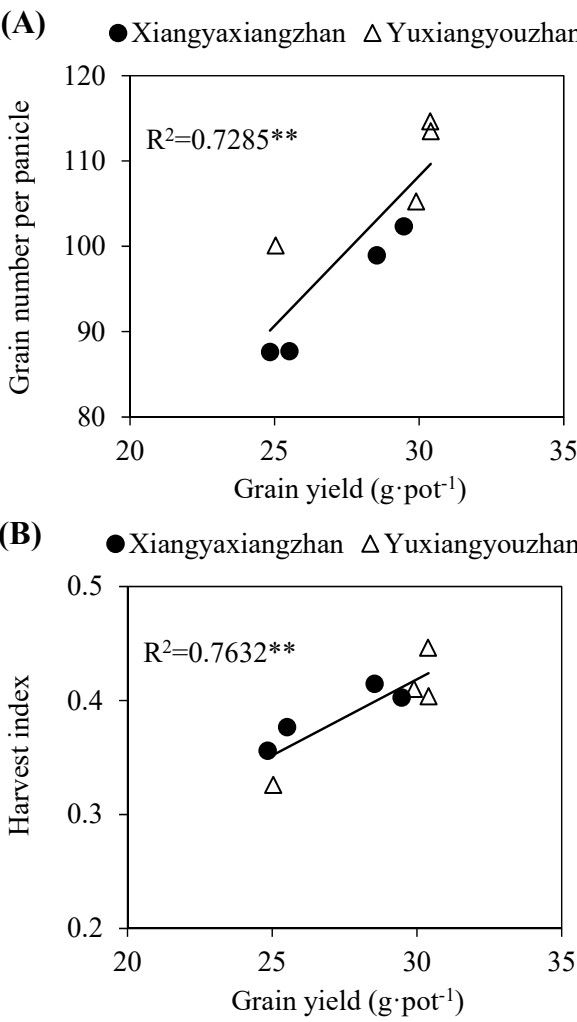

**Figure 7.** Correlation analyses between grain yield and grain number per panicle (**A**) and harvest index (**B**) of fragrant rice. **\*\***, significant at *p* < 0.01 level. 15 d AT, 15 days after treatment; MS, maturity stage.

## 4. Discussion

In the present study, light quality treatments significantly affected the 2AP content in mature grain (Table 1). Previous studies have reported that there may be a 'light quality' effect on the fragrance accumulated in other plants such as strawberry and coriander leaves [33,34]. A previous study has shown that low light-induced 2AP accumulation in fragrant rice [19,20]. As expected, reductions of the 2AP content were detected for all the light quality treatments by 16.67–32.82%. L2 treatment had the greatest effect on 2AP content of Yuxiangyouzhan, which was significantly lower than that of CK. This result further confirmed the negative effect of light on fragrance in fragrant rice. Regarding the experimental varieties, the decrement in 2AP in the Xiangyaxiangzhan was higher than that of Yuxiangyouzhan (Figure 1). The difference between the varieties is also reported by Tu et al. [35] and Okpala et al. [7]. This shows that different varieties have different degrees of sensitivity to light in relation to 2AP. Change in 2AP content at 15 d AT and MS under different light quality treatments for the two varieties was also observed (Table 1, Figure 1). This may be because light can inhibit the 2AP synthesis pathway response from different growth stages. When continuing to provide light treatment until the MS, the varieties gradually adapt to light quality treatments, which results in no significant difference in 2AP content of fragrant rice at the MS. Significant V×T effect on 2AP content in grains was observed (Table 1). The significant interaction effect between variety and treatment indicated that the effect of different light treatments on 2AP content of fragrant

rice varied varieties, which further indicated that different varieties have different degrees of sensitivity to light. Moreover, the crop management or breeding of fragrant rice varieties with different light quality responses in yield and fragrant balance is possible. From the results of previous studies, the 2AP content in grain showed a strong relationship with P5C content [20] and GABA content [19]. In this study, the strong relationships between the 2AP content in grains at MS and the P5C content in grains at 15 d AT ($r^2 = 0.6714$, $p < 0.05$), the P5C content in grains at MS ($r^2 = 0.7846$, $p < 0.01$), and the GABA content in grains at 15 d AT ($r^2 = 0.5222$, $p < 0.05$) were detected (Figure 6). Several previous studies have reported that P5C and proline were precursors of 2AP biosynthesis [11,13,36]. In addition, Poonlaphdecha et al. [5] reported that 2AP accumulation was associated with GABA levels. Moreover, Xie et al. [9,37] and Gao et al. [10] have suggested that GABA application could help improve 2AP accumulation. In this study, the proline, P5C, and GABA content changes in grains and leaves under different light quality treatments (Figures 2–4). These results indicate that light quality treatments which decreased 2AP content in mature grain are highly associated with the reduction of P5C in grain and GABA accumulation in grain at 15 d after treatment. This provides evident that for the presumption that P5C is the precursor of 2AP and indicates that GABA may affect the content of 2AP by affecting the synthesis of P5C.

Previous studies have shown that LED treatment could significantly promote early rice growth, such as at the seedling stage [38,39]. Though there are few studies investigating LED light quality treatment on rice growth at later phases, it could be predicted that light quality treatment during the grain filling period significantly affects grain yield. Different light quality treatments had varying effects on the grain yield of different varieties. Light quality treatment could improve the grain yield, grain number per panicle, filled grain percentage, and harvest index. L3 treatment had the greatest effect on grains yield of Xiangyaxiangzhan, which was significantly higher than that of CK (Tables 1 and 2). Ohashi et al. [40] showed that the biomass production of rice grown under red light supplemented with blue light was higher than that of plants grown under red light alone conditions. In addition, Borowski et al. [41] indicated that red-blue light can increase the stomatal conductance, photosynthesis, and transpiration parameters of lettuce leaves, which were higher than that of single red light or single white light. It seems that combined light quality is more beneficial to the growth and yield formation of rice than single light quality. However, Zheng et al. [42] showed that the yield of cherry tomato under red light was higher than that of composite light and blue light treatment could promote the early turning of tomato color. This suggested that the same plant responds differently under different light quality conditions. For rice, plants at different growth stages may have different requirements for various light qualities. Grain yield is related to grain number per panicle and harvest index (Figure 7), which indicated that light treatments could affect the grain yield by affecting the yield formation and distribution of dry biomass. Moreover, the stem and leaves dry weight of Yuxiangyouzhan under L3 treatment was lower than that of CK (Table 2). This suggested that the photosynthetic product transported from the stem and leaves to the grain was higher during the grain filling period under light quality treatment. Overall, light quality treatment increased the grain yield by regulation of yield-related traits, biomass accumulation, and distribution.

The antioxidant enzymes such as SOD, POD, CAT are important in plant resistance to stress, which is closely related to plant metabolism. Moreover, MDA is the peroxidation product of membrane lipid, and the increase of its content often indicates the decrease of plant resistance. Previous studies have shown that blue light could enhance the antioxidant capacity of strawberries [43]. Zheng et al. [44] reported that, compared to full red light, the combination of red and blue light was more likely to improve the antioxidant capacity of *photinia × fraseri* plantlets in vitro. But there is no study on the effect of light quality on stress resistance physiology of fragrant rice leaves. In this study, L1 treatment was beneficial to the activity of antioxidant enzymes, while L2 treatment only increased the activity of SOD. Compared with CK, MDA content was significantly decreased by L3 treatment

but increased by other treatments (Figure 5). This suggested that supplemental light treatments during the grain filling period could conduce the resistance of fragrant rice, and the cell membrane lipid is damaged. However, L1 treatment could significantly increase the antioxidant capacity of Xiangyaxiangzhan, which is adverse for Yuxiangyouzhan. Moreover, Ueno et al. [45] showed that red light can induce CAT activity in rice, which is different from the present study (Figure 5), which is mainly due to the difference in light density and stages of light treatments conducted. It has been reported that the antioxidant capacity of wheat cultivated under the combination of red and white light is significantly improved compared with pure white light [46]. It is possible that supplemental combinations of light with different ratios during the grain filling period can also modulate the resistance ability of fragrant rice. Furthermore, the difference in plant species, varieties, growth conditions, and sampling stage may lead to changes in antioxidant enzyme activity.

The effect of different light quality treatments on grain quality in fragrant rice has rarely been reported. Mo et al. [19] reported that shading during the filling stage could significantly increase the protein content of fragrant rice. In this study, variety (V), light quality treatment (T), and V×T significantly affected grain quality, except milled rice rate and head rice rate (Table 1). This suggested that different light quality treatments had different effects on the grain quality of different varieties, and the selection of variety should be considered in the process of the practical application of optimization light quality. The light quality treatments decreased chalk rice rate, chalkiness, and protein content, but increased the length-to-width ratio (Table 3). This result showed that the light quality treatments improved the appearance quality and cooking quality of the grains. Further in-depth study into the light quality effect on grain quality formation in fragrant rice is needed.

## 5. Conclusions

Compared with CK, the light quality treatments increased grain yield but decreased 2AP content in mature grains. L3 treatment, in particular, significantly increased the grain yield of Xiangyaxiangzhan, with a smaller reduction in 2AP content in grains than that of other light quality treatments. The light quality treatments decreased chalk-white rice rate, chalkiness degree, and protein content, but increased length-to-width ratio. Xiangyaxiangzhan was more sensitive to light quality treatment than Yuxiangyouzhan. Light quality treatments show that regulation affects yield-related traits, biomass accumulation, antioxidant physiology, and 2AP formation-related physiology. This study confirms the contradiction between yield and fragrance in fragrant rice.

**Author Contributions:** Z.M., data curation; H.X. and W.X., formal analysis; W.X. and X.L., investigation; W.X., methodology; Z.M., supervision; X.T. and Z.M., writing—original draft; H.X., W.X., and X.L., writing—review and editing; S.P., M.D., S.W., H.T., X.T., and Z.M. All authors have read and agreed to the published version of the manuscript.

**Funding:** This research was funded by the National Natural Science Foundation of China, grant number 316601244; the National Natural Science Foundation of China, grant number 31971843.

**Institutional Review Board Statement:** Not applicable.

**Informed Consent Statement:** Not applicable.

**Data Availability Statement:** The data sets supporting the results of this article are included within the article.

**Conflicts of Interest:** The authors declare no conflict of interest.

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
