# Peer review of "Effects of Light Quality Treatments during the Grain Filling Period on Yield, Quality, and Fragrance in Fragrant Rice"

_agronomy, doi:10.3390/agronomy11030531_

Round 1

Reviewer 1 Report

This paper investigates the effects of different LED light treatments during the grain filling period on seed quality, yield and fragrance in jasmine rice. It has previously been shown that low light levels are beneficial for fragrance but higher light levels are preferable for yield in fragrant rice. In this work rice plants (two different cultivars) were exposed to different light treatments during the grain filling period, samples of leaves and grain were harvested and tested for various quality parameters.

My main comment is that many parameters have been investigated, but there is not a great deal of discussion on what the results show. Firstly, there is insufficient information in the introduction on why particular treatments have been used, and why certain results recorded. Some background on light treatments in the introduction would be beneficial; low light levels are known to affect grain fragrance and yield, but why have the treatments of 100% blue light and 100% red light been selected for this work? Is blue/red light associated with shading for example – it would be good to include this information for readers who are not experts in light treatments. There is also insufficient discussion on the interpretation of the results. The concluding paragraph sums up the work by saying that it has been shown that light has a negative effect on fragrance and a positive effect on yield. This result has previously been shown to be true by others, and I don’t think that this finding alone is novel enough to warrant publication. However, many other results are presented in the paper, and if there was further discussion around what some of these results mean then publication could be justified. The discussion could be greatly improved by considering, for example, whether all light treatments had the same effect, or did certain treatments have greater effects than others? Further discussion on differences between the varieties – the light treatment resulted in a decrease in 2AP in both varieties, but the decrease was much greater in Xiangyaxiangzhan, why? Are some varieties are less sensitive to light treatments? There is a significant effect of the interaction between variety and treatment for many of the parameters tested – what are the implications of this?

There are statements made in the discussion that repeat the results, but give no further explanation – L326-328 for example states that “antioxidant enzyme activity in leaves and MDA content at MS in leaves was significantly affected by the variety, light quality treatment and VxT effect”. Is there a reason for this? If further discussion is not possible, then such statements do not really warrant repeating.

The manuscript requires editing to ensure the English is correct throughout.

In addition I have the following comments:

  • The rice plants are referred to as both cultivars and varieties. Use the correct term and be consistent.
  • Ensure that details of replication are made clear in the methods. Figures have error bars, but there is no explanation in the figure caption of how many replicates have been used to calculate these errors.
  • One of the light treatments is repeatedly referred to as being 33% red light, 33% red light 33% white light – e.g. L25, 83, 176, 274. I presume this is an error.
  • L81 - Light treatments were applied from June 10th – June 30th, which corresponded to growth stages R5 to R9. Was this exactly the same for both cultivars? I presume not as one has a longer growth period than the other. Can you comment further on this? Would a slightly different timing of treatment with respect to growth stage have had any impact on the results?
  • Give full details of light treatments – for example how many hours was light treatment supplied for per day?
  • L138 - were grains harvested from six plots, or from six pots?
  • Table 1 needs an explanation of what the values are – are they F Values? Mean Squares…?
  • The caption below Table 1 (L167-172) doesn’t make sense – there are no lower case letters in the table.
  • In Figure 1 not all of the results have error bars. Is this an omission?
  • There are two Figure 6s.
  • L323 – should be stem, not steam.

Reviewer 2 Report

agronomy-1063437

Comments for the authors

The authors investigated an interesting topic but some of its statement is incorrect and the conclusion is too general. The presentation and design of experiments is incomplete it should be improved. The figures, although well edited, are inaccurate in their notes, they should be corrected. Grammar mistakes occurring make some sentence to be unclear. The references in the text and reference list do not meet the requirement of Agronomy journal.

 Detailed comments

Material and methods:

74 line: “seedings” What do you mean by seeding? Do you mean seedlings or planting seeds? The sentence is unclear you should correct it.

-How many repetitions did you use? How many pots were in one repetition and light treatment? These should be presented.

-Why did you use a double 33.3% red light treatment? It should be explained.

Results:

162-165 lines: Fig 1A and Fig.1B should be indicated in the text for appropriate variety and light treatment in order to make the results easier to follow.

167-169 lines: under Table 1 “A lower-case letter…” has to be deleted as neither letters and L1,L2,L3 are found in the table. You only showed the parameters of ANOVA.

174 line: Check the standard error for the Yuxiangyouzhan variety (fig.1B).

- Some suggestions for note of figures: mark the Fig.2,3,4 in bracket A and B at the grain and C and D at the leaves. This will make the presentation of the figures more expressive.

-You should better highlight which light quality treatment increased the yield and which was effective in the production of fragrance in the mature grain.

- Was there a difference in the sensitivity of the varieties?

Other comments

78 line: “loess” can be loss? Control the sentence.

280 line and 307 in discussion:  r2=0.6714 for P5C contain in grain at MS is wrong, the right value is r2=07846 as seen in Fig.6B.

-What was the correlation between 2AP and GAPA content of grain at MS?

292 line: There are two Fig.6. Fig.6 presenting the relationship between grain and harvest index must be corrected to Fig 7. In the comment you could use A and B in parentheses to make it more expressive.

294 line: “AT…..MS” must be deleted.

319 line: “it can predictable…” What did light treatment significantly with the grain yield? Check the sentence.

323 line: This statement is right only for Yuxiangyouzhan variety and L3. Please control it.

333 line: “…Yield related treats..” Please check it.

325 line and conclusion should highlight which light quality treatment has the most beneficial effect on the grain yield and improves the fragrance production in grain. Is there any difference between the varieties?

Round 2

Reviewer 1 Report

An addition has been made to the introduction to provide more background information on the effects of different light treatments, and this is an improvement.

My main comment was that there was insufficient discussion on the interpretation of the results. An effort has been made to include some detail on the differences in response to light treatments between the two varieties, however my other comments have not been addressed (did all light treatments have the same effect, what are the implications of the significant variety and treatment interaction). The authors gave some information about these aspects in their response to my comments, but these comments have not been included in the manuscript discussion. I feel that the discussion has not been sufficiently improved.
